# A Comprehensive Evaluation of Generating a Mobile Traffic Data Scheme without a Coarse-Grained Process Using CSR-GAN [note 2]

**DOI:** 10.3390/s22051930

**Published:** 2022-03-01

**Authors:** Tomoki Tokunaga, Kimihiro Mizutani

**Affiliations:** 1Graduate School of Science and Engineering, Kindai University, 3-4-1 Kowakae, Higashiosaka 577-0818, Osaka, Japan; 2033340424m@kindai.ac.jp; 2Cyber Informatics Research Institute, Kindai University, 3-4-1 Kowakae, Higashiosaka 577-0818, Osaka, Japan

**Keywords:** conditional GAN, SR-GAN, traffic data management

## Abstract

Large-scale mobile traffic data analysis is important for efficiently planning mobile base station deployment plans and public transportation plans. However, the storage costs of preserving mobile traffic data are becoming much higher as traffic increases enormously population density of target areas. To solve this problem, schemes to generate a large amount of mobile traffic data have been proposed. In the state-of-the-art of the schemes, generative adversarial networks (GANs) are used to transform a large amount of traffic data into a coarse-grained representation and generate the original traffic data from the coarse-grained data. However, the scheme still involves a storage cost, since the coarse-grained data must be preserved in order to generate the original traffic data. In this paper, we propose a scheme to generate the mobile traffic data by using conditional-super-resolution GAN (CSR-GAN) without requiring a coarse-grained process. Through experiments using two real traffic data, we assessed the accuracy and the amount of storage data needed. The results show that the proposed scheme, CSR-GAN, can reduce the storage cost by up to 45% compared to the traditional scheme, and can generate the original mobile traffic data with 94% accuracy. We also conducted experiments by changing the architecture of CSR-GAN, and the results show an optimal relationship between the amount of traffic data and the model size.

## 1. Introduction

Mobile traffic data have increased rapidly in recent years due to the dramatic spread of mobile devices and online services. According to Ericsson research, the amount of monthly mobile traffic data is expected to reach 49 EB per month by the end of 2020 and 237 EB by 2026 [1]. To handle a large amount of future mobile traffic data, it is important to analyze the trends of mobile traffic patterns in order to deploy mobile devices and online services on a large scale in urban areas [2,3,4,5]. However, the computational power to achieve this is high [6], since specialized equipment (e.g., measurement probes) is required. Furthermore, the storage cost of mobile traffic data is becoming much higher, since it increases enormously with the size and population density of a target area. In order to solve these problems, schemes to estimate a large amount of traffic data have been proposed. In particular, with the recent advances in neural network technology, many schemes using deep neural networks have been widely proposed.

For example, recurrent neural networks (RNNs) used in LSTM architecture [7] are superior to traditional machine learning approaches in terms of accurate traffic generation. ConvLSTM [8] and spatio-temporal neural network (STN) [9], which are the advanced versions of RNNs, are useful for short-term predictions of mobile traffic data and could also be applied for generating the mobile traffic data. However, these schemes are not suitable for generating long-term mobile traffic data, because they can only forecast the traffic volume for a few tens of hours.

In contrast, ZipNet-GAN [10], the state-of-the-art scheme, can generate long-term mobile traffic data for up to 2 months, since it does not use LSTM. For using ZipNet-GAN, the original data (i.e., raw mobile traffic data) must be transformed to coarse-grained representation data. Then, ZipNet-GAN [10] generates the original traffic data from the coarse-grained data. In ZipNet-GAN, mobile traffic data can be represented in a three-dimensional form (*x*, *y*, *volume*), where *x* and *y* are longitude and latitude, and *volume* is the amount of traffic data at that point. If we preserve the time series of mobile traffic data covered by a rectangular area (e.g., a grid cell or traffic map), the storage cost can be expressed as X·Y·T, where *X*, *Y*, and *T* are the ranges of longitude, latitude, and time. In a coarse-grained representation, the mobile traffic data X·Y·T is transformed to a coarse-grained data (X/C)·(Y/C)·T, where *C* is a constant value. Note that the original paper proposed ZipNet-GAN used a maximum of C=10. From the coarse-grained data (X/C)·(Y/C)·T, ZipNet-GAN generated the original traffic data X·Y·T. As a result, the original traffic data can be generated accurately, but the storage cost for coarse-grained data increases as the number of the rectangular traffic data area and the time span increase.

To solve the storage cost problem, we propose a scheme to generate mobile traffic data by using CSR-GAN without coarse-grained data. Our proposed scheme generates the original traffic data from the random numbers and the target time data (label). With this scheme, the storage cost of the mobile traffic data X·Y·T can be reduced to only *T* time labels, as random numbers are temporarily used and do not need to be preserved. In other words, our scheme need not preserve coarse-grained data, so it can generate mobile traffic data more efficiently than ZipNet-GAN. The basic concept of this scheme was published in [11], but it lacked both exhaustive verification and a suitable neural network model hitherto. In this paper, we address both problems and clarify the effectiveness of our proposed scheme from several perspectives.

## 2. Related Work

In this section, we describe the works related to our research, and discuss the differences.

The necessity of mobile traffic data analysis and its cost: Reference [5] mentions the necessity of the techniques to accurately infer mobile traffic data analysis to deploy the future 5G systems and large-scale online services. In addition, it focuses on the versatility of the deep learning and discusses how it can be applied to mobile traffic data analysis.

Reference [6] surveyed the traffic monitoring systems used in the conventional mobile traffic analysis schemes. This shows the need for RNC probes to record the fine-grained state changes of each mobile device, and points out the problem of incurring considerable storage costs for preserving the state’s information. In addition, PGW probes, which cover a wide geographical area compared with RNC probes, can monitor an increase in mobile traffic with small overheads; however, the storage cost problem cannot be resolved.

A scheme for generating mobile traffic data: Reference [8] aimed to forecast the multi-service mobile traffic using both the sequence-to-sequence (S2S) learning paradigm and convolutional long short-term memory (ConvLSTM). Its scheme effectively extracts complex spatio-temporal features of mobile traffic data, and can forecast the future demand for individual services with high accuracy, even if the service scale becomes larger. In an experiment using real-world mobile traffic data, they demonstrated that the proposed scheme can forecast the mobile traffic volume generated by dozens of different services. The inputs for ConvLSTM are 1-h traffic data, and the outputs are future hourly traffic data. This scheme focuses on the prediction toward the future hour, and discards the storage cost for preserving past traffic data.

Reference [9] proposed a scheme for fine-tuning the STN, which realizes generating accurately mobile traffic data with lower mobile traffic data. Furthermore, they proposed the Double STN scheme (D-STN), which enables long-term prediction of mobile traffic data by uniquely combining STN predictions with historical statistics. In experiments using a real-world mobile traffic dataset collected over 60 days in both urban and rural areas, the proposed (D-STN) scheme achieved a long-term prediction of up to 10 h with high accuracy. This scheme also takes many input data for estimations. Reference [12] also focuses on forecasting the future mobile traffic data in urban areas. They used various network oscillation factors, such as jitter, packet loss, and delay (i.e., call detail record (CDR)) for enhancing the forecast ability; in addition, they adopted LSTM architecture [7] for realizing long term forecasting. Reference [13] proposed adaptive multi-receptive field spatial-temporal graph convolutional networks (AMF-STGCN), which is upward compatible with graph convolutional networks (GCNs) and can forecast future mobile traffic data with high accuracy. Its neural network model is specific to forecast the spatio-temporal features of the mobile traffic data. Both studies had the same motivation for forecasting future mobile traffic data; however, their methods require massive amounts of past traffic data as inputs. For example, the model in [13] takes the past 6 days’ traffic data to forecast the next 6 days’ traffic data.

Reference [10] proposed ZipNet-GAN, which is a combination of the zipper network (ZipNet) model and the generative adversarial neural network (GAN), which generates the original mobile traffic data from the coarse-grained data corresponding to the original mobile traffic data. In experiments using a real-world mobile traffic dataset, they demonstrated that ZipNet-GAN can forecast mobile traffic data with high accuracy and up to 100 times granularity. This is the first time that the concept of super-resolution has been applied to large-scale mobile traffic analysis. While realizing the high accuracy of mobile traffic generation, the coarse-grained data must be preserved, and their storage cost is large compared to that of our scheme.

## 3. Proposed Scheme: Conditional-Super-Resolution GAN

### 3.1. Essential Background of GANs and ZipNet-GAN

The GAN [14] is an unsupervised learning framework for generating artificial data from random numbers. In general, GANs consist of two neural network models, a generator *G* generating target artificial data from random numbers, and a discriminator *D* discriminating whether the output of *G* is derived from real training data or artificial data. In a training process of GANs, a generator *G* takes random data seed (e.g., Gaussian or uniformly distributed) *z*~Pn(z), and generates an output G(z), which aims to follow a target unknown data distribution (pixels in images, traffic data, etc.). The discriminator *D* randomly picks the data generated by *G* (i.e., G(z)) and other data sampled from the target data distribution (i.e., *x*), and is trained to maximize the probabilities of correctly identifying fake and real data. *G* is trained to generate data whose distribution is as close as possible to *x* for maximizing the probability that *D* makes mistakes. This interaction looks like a mini-max two-player game where each model is trained separately while fixing the other one. Summarizing these processes, the GAN aims to solve the following mini-max problem:(1)$$\min_{G}\max_{D}V\left(G,D\right)=E_{x∼Pr(x)}\left[logD\left(x\right)\right]+E_{z∼P_{n}\left(z\right)}\left[log\left(1-D\left(G\left(z\right)\right)\right)\right]$$
where $E_{x∼Pr(x)}\left[logD\left(x\right)\right]$ shows the expected value of logD(x) when the target data distribution is input to *D*. $E_{z∼P_{n}\left(z\right)}\left[log\left(1-D\left(G\left(z\right)\right)\right)\right]$ also shows the expected value of log(1−D(G(z))) when the data distribution generated by *G* is input to *D*.

Once trained, the generator *G* can generate artificial data samples from the target distribution given noisy inputs (random numbers) *z*, and its output data are closer to the target training data. Note that we need not preserve the discriminator *D* because it is not needed after completing generator learning.

In ZipNet-GAN, generator *G* is used to generate the original traffic data from the coarse-grained data, and discriminator *D* is used to correctly judge whether the data are original traffic data or generated traffic data. Figure 1 shows an abstraction of ZipNet-GAN for generating mobile traffic data. First of all, we generate coarse-grained data from the original traffic data. Recall that the original mobile traffic data X·Y·T are transformed to a coarse-grained data (X/C)·(Y/C)·T, where *X*, *Y*, *T*, and *C* are longitude, latitude, time, and a constant value. Second, generator *G* generates traffic data from the coarse-grained data. Next, discriminator *D* takes a mixture of both the original traffic data and the traffic data generated by *G* as input data, and it judges whether each input is an original datum or not. By repeating this training process, ZipNet-GAN’s generator *G* can generate the original mobile traffic data from only coarse-grained data. Therefore, the storage cost of mobile traffic data in ZipNet-GAN comes from preserving the coarse-grained data (i.e., (X/C)·(Y/C)·T) and generator model (i.e., *G*). Note that we need not preserve the discriminator *D* because it is not needed after completing generator learning.

### 3.2. Conditional-Super-Resolution GAN (CSR-GAN)

In this paper, we propose a scheme using CSR-GAN to generate a large amount of traffic data mobile traffic data without requiring a coarse-grained process. Figure 2 shows an abstraction of the CSR-GAN in our proposed scheme. CSR-GAN is a type of GAN that is composed of a combination of conditional GAN (C-GAN) [15] and super-resolution GAN (SR-GAN) [16]. C-GAN is a scheme of generating original data from a certain label that could be any kind of auxiliary information, such as class labels or time data. We can perform the conditioning by feeding the label into GAN. In contrast, SR-GAN has the same features as ZipNet-GAN, but it outperforms ZipNet-GAN in terms of generating accurate original data. Therefore, CSR-GAN, combining the two, can generate mobile traffic data from random numbers and target time data (label) directly. Thus, the storage cost is only the sizes of the list of target time data (label). The formula for the CSR-GAN is shown below.
(2)$$\min_{G}\max_{D}V\left(G,D\right)=E_{x∼Pr(x)}\left[logD\left(x|L\right)\right]+E_{z∼P_{n}\left(z\right)}\left[log\left(1-D\left(G\left(z|L\right)\right)\right)\right]$$

In CSR-GAN, the generator *G* takes random number seeds (e.g., Gaussian or uniformly distributed) *z*~Pn(z) with a target time label *L*, and generates an output $\hat{x}$ that aims to follow a target mobile traffic data. In contrast, the discriminator *D* randomly picks the data generated by *G* (i.e., $\hat{x}$~G(z|L)), and others sampled from the target data distribution (i.e., *x*~$P_{r}\left(x\right)$) with the label *L*, and makes judgments from them.

In detail, *G* is trained to generate data whose distribution is as close as possible to $P_{r}\left(x\right)$ for maximizing the probability that *D* makes mistakes. In contrast, *D* is trained to maximize the probabilities of correctly judging whether $\hat{x}$ is fake or *x* is real. Note that x|L and z|L denote the conditional value *L* added to *x* and *z*.

Once trained, the generator *G* can generate traffic data from random numbers (noisy inputs) *z* with the target time label, and its output is closer to the target traffic data. Applying the label for GANs architecture enables the generator *G* to output the original traffic data from only random numbers *z* and target time label *L*.

### 3.3. The Implementation of CSR-GAN

The generator *G* of CSR-GAN mainly consists of three components: Conv2Dblock, ResNetBlock, and Pixel-ShufflerBlock. Conv2DBlock consists of Convolution2D layers [17] each containing an activation function layer Parametric RelU (PReLU) [18]. Using Conv2DBlock multiple times, the generator can capture the features of mobile traffic data. ResNetBlock [19] consists of three layers: a Convolution2D layer, a BatchNorm2D layer [20], and an activation function layer PLeLU. This ResNetBlock captures the different features in detail and generates the input for the next convolution layer. Finally, Pixel-ShufflerBlock consists of three layers: the Convolution2D layer, the pixel shuffler layer [21], and the PReLU layer. The pixel shuffler layer expands the input data calculated by the above-mentioned layers, which aims to up-sample the input to the desired output. Pixel-ShufflerBlock plays an important role in extracting the specific data distribution and temporal features of mobile traffic data. Using these architectures, the generator *G* can generate the original traffic data from input data consisting of both normal random numbers and target time label. For tuning this generator, we also use mean squared error (MSE) for the loss function of the generator (The details are in Section 4) and minimize its loss for accurate data generation.

On the other hand, the discriminator *D* consists of Conv2D, BatchNorm2D, and a leaky Relu (LReLU) [22], such as the VGG network [23]. The reason why we adopt LReLU is that the non-linearity fitting accuracy of the neural network models can be improved. We also set the loss function of discriminator *D* as binary cross-entropy (BCE), since the purpose of the discriminator is to solve a binary classification problem: whether the traffic data are real or not. Discriminator *D* takes as input data a mixture of both the original traffic data and the traffic data generated by generator *G*, and it judges whether each input is original data or not correctly. The outputs of the discriminator are [0,1] values. In the learning phase of the CSR-GAN, the discriminator *D* tries to minimize the error in the judgments. In contrast, the generator *G* not only tries to minimize the error between the output data and the original traffic data, but also maximizes the discriminator’s output error. The combination of maximizing and minimizing errors during the learning phase contributes to a highly accurate generator output. By using CSR-GAN, the mobile traffic data can be generated from only the normal random numbers and the target time data (label). Therefore, the storage cost of traffic data reduces to just storing the number of time labels.

### 3.4. Algorithm for Training CSR-GAN

To train the CSR-GAN, we employ Algorithm 1. The purpose of this training is to find out the optimal parameters of the neural network components, *G* and *D*.

The iteration block concerning the generator’s training is in lines 2–6. At first, the generator calculates the gradient that powered the MSE between the ground truth (original traffic data) and the generator’s output, and the probability of correct judgment of the discriminator. With the gradient, our proposed scheme updates the CSR-GAN model through the Adam optimizer [24].

The iteration block concerning the discriminator’s training is in lines 7–10. The BCE of the discriminator is calculated with the time label and the ground truth data. With the BCE, the CSR-GAN model is tuned for minimizing the error through Adam optimizer.

Repeating this iteration in $n_{D}$ and $n_{G}$ times, respectively, the training of both generator and discriminator is completed. Especially in the initial stage of both training processes, *D* can easily discriminate the data generated by *G*. To solve this problem, in this study, we initialized *G* by minimizing MSE until convergence, similarly to ZipNet-GAN. We also set $n_{G}$ and $n_{D}$ to 1 and the learning rate λ to 0.0001.

Next we discuss the dataset we employed to train and evaluate the performance of the proposed CSR-GAN.
**Algorithm 1** The CSR-GAN training algorithm. **Inputs:**   random numbers *z* (e.g., Gaussian or uniformly distributed),   target time label *L*, learning rate λ,   generator and discriminator sub-epochs,$n_{G}\;\mathrm{and}\;n_{D}$,   dataset *M*. **Initialize:**   Generative and discriminative models, *G* and *D*,   parameterised by $\Theta_{G}$ and $\Theta_{D}$.1: **while**
$\;\Theta_{G}\;and\;\Theta_{D}\;not\;converge$
**do**2: **for**
$e_{G}=1\;to\;n_{G}\;$
**do**3:   sample random number *z*, target time label *L* and ground truth *x*.4:   $g_{G}\leftarrow \Delta_{\Theta_{G}}[\frac{1}{\left|M\right|}\sum_{i=1}^{\left|M\right|}\left(1-\log D\left(G\left(z|L\right)\right)\right)\cdot ||x-{G\left(z|L\right)||}^{2}$.5:   $\Theta_{G}\leftarrow \Theta_{G}-\lambda \cdot Adam\left(\Theta_{G},g_{G}\right)$.6: **end for**7: **for**
$e_{D}=1\;to\;n_{D}\;$
**do**8:   sample target time label *L* and ground truth *x*.9:   $g_{D}\leftarrow \Delta_{\Theta_{D}}\left[\frac{1}{\left|M\right|}\sum_{i=1}^{\left|M\right|}\log D\left(x|L\right)+\frac{1}{\left|M\right|}\sum_{i=1}^{\left|M\right|}\log\left(1-D\left(G\left(z|L\right)\right)\right)\right]$.10:  $\Theta_{D}\leftarrow \Theta_{D}+\lambda \cdot Adam\left(\Theta_{D},g_{D}\right)$.11: **end for**12: **end while**

## 4. Evaluation

The dataset we used for CSR-GAN evaluation consisted of real mobile traffic data collected from Telecom Italia’s Big Data Challenge [25]. Each record in the dataset contains the cellular traffic volume from every 10 min at Milan and Trentino. The measurement period was between 1 November 2013 and 1 January 2014 (2 months) notes that we used all the published data. This was obtained by combining call detail records (CDR) generated by the user interactions on base stations that the user interactions indicate, namely, each time a user started/ended the internet connection or consumed more than 5 MB. The dataset contains 8928 snapshots of fine-grained mobile traffic measurements in each city. Note that we merged each dataset into one file for this evaluation, and the 8928 snapshots represent a set of one snapshot every 10 min over 62 days (when we used both cities’ datasets, the number of snapshots was 17,856). The coverage area of the mobile traffic data in each city is divided into 100 × 100 or 117 × 98 squares, with each point on the map indicating the traffic volume. Note that 117 × 98 was transformed to 100 × 100 in CSR-GAN.

We implemented our proposed scheme by using Python3 and the PyTorch libraries [26]. We changed the input elements to carry out an exhaustive verification of our scheme. In previous sections, we explained that the input data for the generator consists of two elements: the target time label and random numbers. Here, we changed the number of random numbers from 10 to 100 to 1000. Next, we used the four types of representation of the time labels: decimal time, binary time, ordered decimal time, and ordered binary time. Table 1 shows concrete examples of the four types of expression. The decimal time corresponds to the simple time representation consisting of a year, month, day, hour, and minute. Binary time is just the binarized representation of the decimal time. Ordered decimal time indicates the number of data from the first period in all datasets. For example, when the range of the traffic dataset is [201311010000, 201401012350], the ordered decimal time corresponding to 20131101000 is 1. Ordered binary time is just the binarized form of ordered decimal time. The reason why we use the binary notation label is that the neural network model can observe the relationship between input and its output easily in general as the size of input label size becomes larger. With the label expression, we aimed to improve the learning speed and the accuracy of data generation.

Finally, we adapted MSE and peak signal-to-noise ratio (PSNR) to evaluate the proposed CSR-GAN scheme. We defined the MSE as the average of the error between the ground truth and the outputs by a generator. Note that MSE is already used to minimize the error between output data and original mobile traffic data. The MSE used in this evaluation is defined by the following formula.
(3)$$\mathrm{MSE}=\frac{1}{T}\sum_{k=1}^{n}\left|\right|\left(x_{t}-{\hat{x}}_{t}\right){)||}^{2}$$
where *T*, $x_{t}$, and ${\hat{x}}_{t}$ are the number of snapshots, original traffic data, and generator’s output data at time *t*, respectively. Note that each snapshot corresponds to each time, hence, the time *t*’s snapshot is the same expression as time *t*’s traffic data.

PSNR is a measure of image reconstruction’s quality, which is expressed by the following formula.
(4)$$\mathrm{PSNR}=20\;log\left(Max\;h_{t}\right)-10\;log\left(\mathrm{MSE}\right)$$
where $max\;h_{t}$ is the maximum value of traffic data volume at time *t*. As the quality of the image becomes higher, the PSNR value also becomes higher.

Using these settings, we trained the CSR-GAN to generate the original 100 × 100 traffic volume map from random numbers and the target time labels. The number of training iterations (epochs) was 400, and we set the parameters concerning learning efficiency to the recommended values of the PyTorch libraries. This evaluation was used on SR-GAN for a comparison, which is fully upward compatible with ZipNet-GAN. Note that SR-GAN’s generator generated the original traffic data from coarse-grained data consisting of 10 × 10 traffic data maps.

Figure 3 shows the MSE between CSR-GAN’s generator’s output in the proposed scheme and the original traffic data. In the CSR-GAN, we changed each random number and label representation against the number of epochs. The result shows that the convergence speed was improved, as the number of random numbers decreased (left of Figure 3). By fixing the number of random numbers to 10, we measured the convergence speeds of each input label. The convergence speed was different for the types of target time label representations (right of Figure 3). In detail, the ordered decimal time labels gave unstable results due to the small number of elements in the input vectors compared with binary notation labels. From this fact, we concluded that the binary notation label is more suitable for label representation. Although SR-GAN takes coarse-grained data 10 × 10 as input, there was no difference in the value of MSE convergence between our proposed scheme (CSR-GAN) and SR-GAN. Note that our scheme reduced the input data size significantly compared with SR-GAN.

Next, we conducted the experiments to observe the relationship between the CSR-GAN model size and its performance. In detail, we aimed to find reasonable neural network settings for the target dataset. The target dataset is a mixture of Milan (2 months) and Trentino (2 months) data; hence, we used 4 months of mobile traffic data for this experiment. The reason why we used its dataset is that we wanted to find a reasonable model for a complex dataset. For the target dataset, we used the five model sizes no ResNet, and 1-layer, 2-layer, 4-layer, and 8-layer ResNet, which were applied to ResNetBlock in CSR-GAN. Figure 4 shows the MSE and PSNR against each CSR-GAN model. The results show that there was no difference in the convergence speed in relation to the MSE and PSNR among the different CSR-GAN models. In particular, the convergence speed with no ResNet was less stable than that with ResNet (1-layer, 2-layer, 4-layer, or 8-layer ResNet). As a result, SR-GAN with coarse-grained data as input converged the MSE faster. However, at the end of training (400 epochs), all schemes converged to approximately the same value for all conditions. This result indicates that the proposed scheme achieves the same accuracy as SR-GAN while reducing the storage cost of the input.

Next, Figure 5 shows results about the order of traffic volume and MSE in the distinctive area. We sampled only the traffic volumes from the distinctive areas (i.e., high traffic demand space) and compare their traffic volume’s order and the MSE. The left of Figure 5 shows the graph of the top 100 traffic data volumes sorted in descending order for different sizes of ResNet. The results show that the larger size of the ResNet used, the closer to the ground truth the generated traffic volume. The right of Figure 5 shows the MSE for the top 10, 100, and 1000 and the ground truth. Note that the traffic volumes are sorted in descending order. The result shows that with more than four layers, ResNet can generate accurate traffic volume similar to SR-GAN.

The results show that with more than four layers, ResNet can generate accurate traffic volume similar to the SR-GAN. Next, Figure 6 shows the snapshots of the output data of our proposed scheme with changes in ResNet size and SR-GAN (coarse-grained data as an input). It indicates the proposed scheme with no ResNet or 1-layer ResNet cannot generate traffic volume similar to the ground truth. Focusing on the green circle of the figure, we can see that the characteristics of the ground truth were lost in our schemes. In contrast, our schemes equipped with 4 layers and 8 layers in ResNet successfully captured the characteristics of the ground Truth, accurately. This indicates that using multiple ResNet has a significant impact on the accuracy of the proposed scheme. Next, Figure 7 shows the snapshots of output data of our proposed scheme and SR-GAN (coarse-grained data as an input) at another time. It indicates that the proposed scheme can generate traffic data similar to the ground truth, even if the traffic volume map is sparse. It also indicates that the lower traffic volume map is not affected by the size of ResNet. We observed that our proposed scheme delivers remarkably accurate results, as the texture and details were almost perfectly recovered. In detail, our proposed scheme achieved 94% accuracy, calculated by dividing the sum of each cell of the original traffic data and the data generated by the proposed method. This result is enough to keep up with the results of previous works. With its high accuracy, our proposed scheme does not need a large input dataset as other models do [8,9,10]; therefore, we can conclude that our scheme is a reasonable solution for storing the large mobile traffic data.

## 5. Discussion

In the results, we focused on the accuracy of the proposed scheme’s output. Finally, we compared the storage costs for ZIP compression, SR-GAN, and our proposed scheme. The size of original mobile traffic data measured in Milan was 1.64 GB. After applying ZIP compression for original traffic data, the volume was reduced to 719 MB. Then, the traffic volume after the coarse-grained process was 16.4 MB. However, more storage is also required for the generator model, 29 MB, as SR-GAN must preserve its generator model for generating coarse-grained data form original data. Therefore, the storage size for SR-GAN was 45.4 MB. On the other hand, our scheme needs storage only for the generator model and the time labels for generating the original traffic data. Then, the storage size needed for our data is the same as SR-GAN, and the data of the time labels totals 200 KB; therefore, the storage size is only 29.2 MB. The results show that our scheme can reduce the storage cost by more than 35% compared to SR-GAN. Next, we also evaluated the storage cost for the larger dataset combining Milan and Trentino, whose size was 2.68 GB. The storage size for applying ZIP-compression to the original data was 1.19 GB, and SR-GAN used 55 MB, which contained the coarse-grained data and the generator’s model. On the other hand, our scheme required 29.4 MB. This result indicates that our scheme can reduce the storage cost of mobile traffic data by more than 45% compared to SR-GAN. However, the total amount of data was not reduced significantly, since the targeted raw data were few for both comparison schemes. To investigate the difference in data compression abilities in both schemes in detail, evaluation with larger data set will be needed.

## 6. Conclusions

Mobile traffic data have increased rapidly in recent years due to the dramatic spread of mobile devices and online services. Additionally, the associated storage cost has grown quickly. ZipNet-GAN, a state-of-the-art scheme, can solve the storage cost problem by generating traffic data from coarse-grained data, but the coarse-grained data must be preserved. We proposed here a scheme for generating mobile traffic data without requiring coarse-grained data. Our proposed scheme consists of generative adversarial networks (GANs; CSR-GAN), combining C-GAN and SR-GAN. The proposed CSR-GAN can generate the original traffic data correctly, and the storage cost is relatively modest. In addition, its algorithm is based on two GANs that can be implemented easily, and the convergence time of the model during training is similar to that of Zipnet-GAN when completion is based on PSNR and MSE. In our evaluation, our proposed scheme could reduce the storage cost for real mobile traffic data (i.e., Milan and Trentino) by more than 35% compared to ZipNet-GAN while providing accurate traffic data generation. In addition, CSR-GAN could continue with accurate traffic generation even if the model size was shrunk, and catch a characteristic of mobile traffic, such as a spike feature.

## Figures and Tables

**Figure 1 sensors-22-01930-f001:**
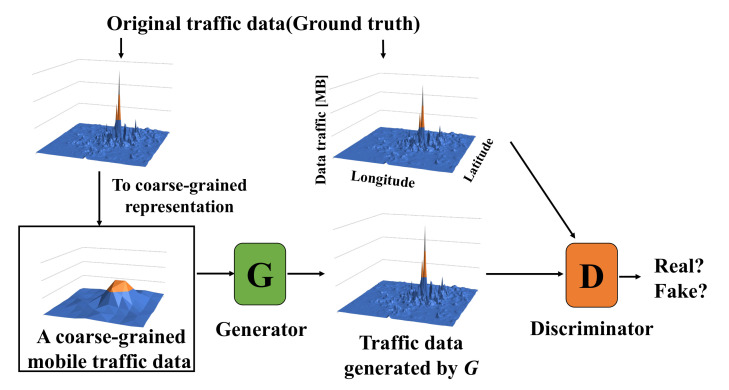
The abstraction of a scheme of generating mobile traffic data by using ZipNet-GAN.

**Figure 2 sensors-22-01930-f002:**
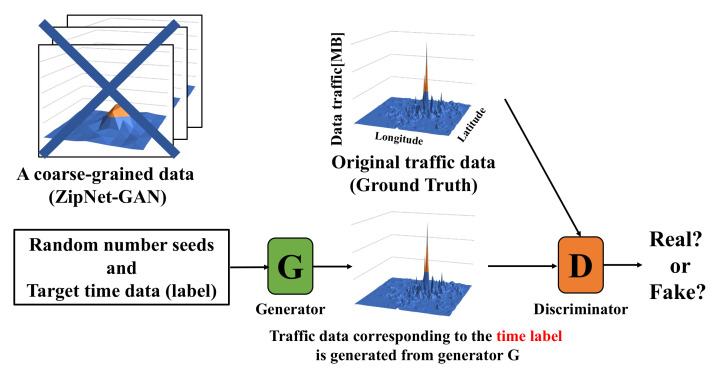
The abstraction of a scheme for generating mobile traffic data by using CSR-GAN (proposed scheme). Our scheme aims to generate the real dataset without coarse-grained data, whereas ZipNet-GAN uses the coarse-grained data.

**Figure 3 sensors-22-01930-f003:**
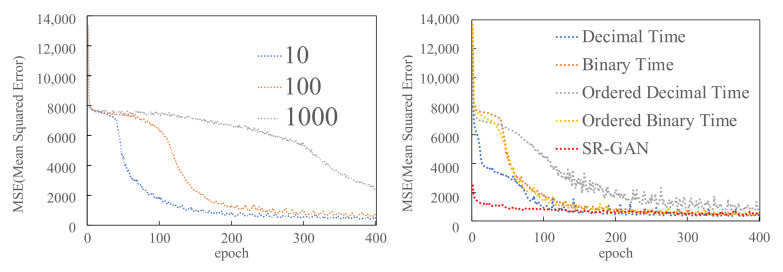
MSE between the generator’s output in the proposed scheme and the original traffic data for each random number and label representation against the number of epochs.

**Figure 4 sensors-22-01930-f004:**
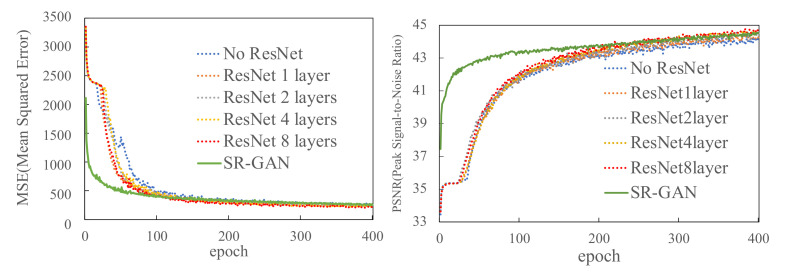
MSE and PSNR between the generator output in our proposed scheme with changes in ResNet size and SR-GAN (input is coarse-grained data).

**Figure 5 sensors-22-01930-f005:**
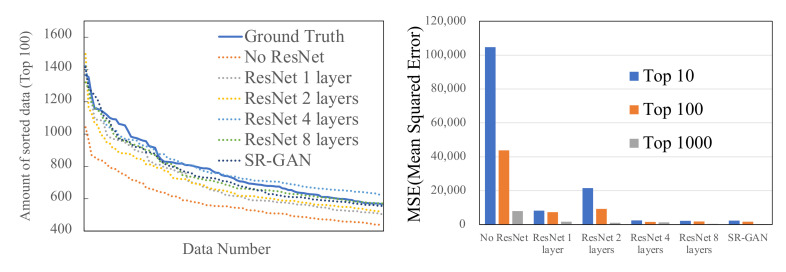
Top 100 traffic data volume scores sorted by volume, and the top 10, 100, and 1000 MSE.

**Figure 6 sensors-22-01930-f006:**
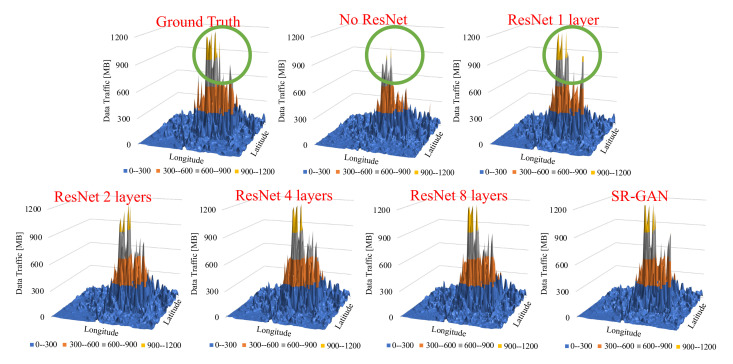
The snapshots of the output data of our proposed scheme with changes in ResNet size and SR-GAN (coarse-grained data as input), using data collected in Milan on 1 November 2013.

**Figure 7 sensors-22-01930-f007:**
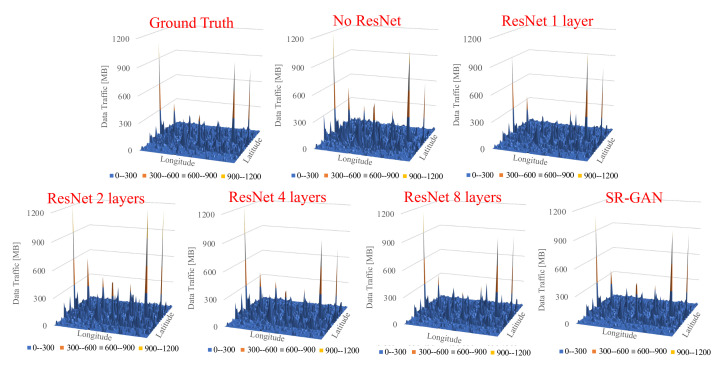
The snapshots of the output data of our proposed scheme with changes in ResNet size and SR-GAN (coarse-grained data as input), using data collected in Trentino on 1 December 2013.

**Table 1 sensors-22-01930-t001:** Four types of representations of the time labels.

Decimal Time	Binary Time	Ordered Decimal Time	Ordered Binary Time
201311010000	10111 ⋯ 10000	0001	00000000000001
201311010010	10111 ⋯ 11010	0002	00000000000010
⋮	⋮	⋮	⋮
201401012340	10111 ⋯ 10100	8927	10001011011111
201401012350	10111 ⋯ 11110	8928	10001011100000

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
