# Peer review of "A Comprehensive Evaluation of Generating a Mobile Traffic Data Scheme without a Coarse-Grained Process Using CSR-GANâ€"

_sensors, 2022, doi:10.3390/s22051930_

Round 1
Reviewer 1 Report
This paper proposed a scheme to generate the mobile traffic data by using CSR-GAN without requiring a coarse-grained process. The autors using experiments using two real traffic data, and validated the accuracy and the amount of storage data.
Suggestions:
The state of the art I would like to see changed after the introduction. Also, add two or three citations that contribute to the related works.
In the evaluation the data set could justify 2 months. Why not more months?
Apart from the results shown, I consider it important to add a discussion section.
Reviewer 2 Report
The authors in their paper proposed a scheme for generating mobile traffic data without requiring coarse-grained data. authors claim that the proposed scheme can reduce the storage cost and can generate the original mobile traffic data with high accuracy. I can recommend publishing the paper after addressing the following notes.
-In the abstract, you say that "can generate the original mobile traffic data with 94% accuracy" but on the results and discussion part I didn't find anything about this number, where did you get 94% please explain it more and it will be suitable to give a detailed comparison between your work and the previous works and not only stated in the related works.
-In the paper "[23]Barlacchi, G.; Nadai, M.D; Larcher, R.; Casella, A.; Chitic, C.; Torrisi, G.; Antonelli, F.; Vespignani, A.; Pentland, A.; Lepri, B. A multi-source dataset of urban life in the city of Milan and the province of Trentino. Scientific Data, Nature, 2015." speak about the case of Milan only where did you get the data for Toronto as you said in your paper "at Milan and Toronto ", also are the samples were taken only between specific hours in the day or overall the day hours.
-here "the areas where the traffic volume is concentrated" do you mean where the traffic volume is higher over space (zones with high traffic demand) or over time ( rush hours )?
- some abbreviations are presented without explaining it (e.g, in the title you wrote the CSR-GAN (Cascaded Super-Resolution Generative Adversarial Network) but after that one cannot find the meaning of CSR in the text) not all the readers can understand the meaning of those abbreviations.
Reviewer 3 Report
This study proposes a data generation approach with mobile traffic data because of its characteristics. Especially the authors emphasize the storage cost on the mobile data. Was the objective of this manuscript developing a data generation approach to reduce data size to decrease the storage data? To me, this manuscript does not include the objective of it. Second, this is not audience-oriented manuscript with the following reasons. 1) related works were included fairly later in the manuscript. This bothers me to get the point why this manuscript needed on the related scientific field. 2) explanations of the methods were broader. 3) the figures including this manuscript do not convey useful information why this can make decreasing the storage cost. The results may show the proposed approach is a suitable methodology to generate randomly with sample data. If then, the authors should change objective and directions on this manuscript. Mobile big data has strength when the data are used with population scale. Therefore, the authors discuss available applications on the data.
Round 2
Reviewer 2 Report
The paper has been improved, however, what is the difference between Toronto and Trentino, in the paper [25] they speak about Trentino (see the title of the paper "A multi-source dataset of urban life in the city of Milan and the Province of Trentino") and not Toronto ( which is a city in Canada) and I am still asking where did you get the data of the Toronto? maybe you should replace Toronto with the province of Trentino ( that coincides with the paper [25] where the data came from Italy (both Milan and Trentino in Italy) ).
Author Response
Thank you for reviewing our paper.
As the reviewer pointed out, the city name was wrong. So, we modified the city name of Toronto to Trentino.